# The Novel *gem*-Dihydroperoxide 12AC3O Suppresses High Phosphate-Induced Calcification via Antioxidant Effects in p53LMAco1 Smooth Muscle Cells

**DOI:** 10.3390/ijms21134628

**Published:** 2020-06-29

**Authors:** Naoko Takase, Masatoshi Inden, Shunsuke Hirai, Yumeka Yamada, Hisaka Kurita, Mitsumi Takeda, Eiji Yamaguchi, Akichika Itoh, Isao Hozumi

**Affiliations:** 1Laboratory of Medical Therapeutics and Molecular Therapeutics, Gifu Pharmaceutical University, 1-25-4 Daigaku-nishi, 1-1-1 Gifu 501-1196, Japan; 126039@gifu-pu.ac.jp (N.T.); inden@gifu-pu.ac.jp (M.I.); 126027@gifu-pu.ac.jp (S.H.); 155076@gifu-pu.ac.jp (Y.Y.); kurita@gifu-pu.ac.jp (H.K.); 2Laboratory of Pharmaceuticals Synthetic Chemistry, Gifu Pharmaceutical University, 1-25-4 Daigaku-nishi, 1-1-1 Gifu 501-1196, Japan; 156023@gifu-pu.ac.jp (M.T.); yamaguchi@gifu-pu.ac.jp (E.Y.); itoha@gifu-pu.ac.jp (A.I.)

**Keywords:** phosphate, reactive oxygen species, calcification, dihydroperoxides

## Abstract

The excessive intake of phosphate (Pi), or chronic kidney disease (CKD), can cause hyperphosphatemia and eventually lead to ectopic calcification, resulting in cerebrovascular diseases. It has been reported that reactive oxygen species (ROS), induced by high concentrations of Pi loading, play a key role in vascular calcification. Therefore, ROS suppression may be a useful treatment strategy for vascular calcification. 12AC3O is a newly synthesized *gem*-dihydroperoxide (DHP) that has potent antioxidant effects. In the present study, we investigated whether 12AC3O inhibited vascular calcification via its antioxidative capacity. To examine whether 12AC3O prevents vascular calcification under high Pi conditions, we performed Alizarin red and von Kossa staining, using the mouse aortic smooth muscle cell line p53LMAco1. Additionally, the effect of 12AC3O against oxidative stress, induced by high concentrations of Pi loading, was investigated using redox- sensitive dyes. Further, the direct trapping effect of 12AC3O on reactive oxygen species (ROS) was investigated by ESR analysis. Although high concentrations of Pi loading exacerbated vascular smooth muscle calcification, calcium deposition was suppressed by the treatment of both antioxidants and 12AC3O, suggesting that the suppression of ROS may be a candidate therapeutic approach for treating vascular calcification induced by high concentrations of Pi loading. Importantly, 12AC3O also attenuated oxidative stress. Furthermore, 12AC3O directly trapped superoxide anion and hydroxyl radical. These results suggest that ROS are closely involved in high concentrations of Pi-induced vascular calcification and that 12AC3O inhibits vascular calcification by directly trapping ROS.

## 1. Introduction

In recent years, the accelerating spread of processed foods and a Westernized diet have caused concerns regarding excessive phosphoric acid intake in Japan. Inorganic phosphate (Pi) functions as an energy transporter and is an essential building block of cell membranes and bones. The excessive intake of Pi, or chronic kidney disease (CKD), leads to hyperphosphatemia that eventually causes ectopic calcification [1,2]. Cardiovascular disease, due to vascular calcification, is one of the leading causes of death among CKD patients [3,4]. Cardiovascular events, including vascular calcification, account for 50% of all deaths in patients with CKD [3,5]. Hyperphosphatemia, which is a common CKD symptom, is a risk factor for arterial calcification, and serum Pi levels correlate with cardiovascular morbidity and mortality in CKD patients [6,7]. Although the kidney is a common site of ectopic calcification in CKD, neurodegenerative diseases, in which ectopic calcification occurs in the brain, have also been reported. Idiopathic basal ganglia calcification (IBGC)—also known as Fahr’s disease or, more recently, primary familial brain calcification (PFBC)—is a rare neuropsychiatric disease characterized by ectopic bilateral calcifications, mainly in the basal ganglia, but also the cerebellum, brain stem, and subcortical white matter [8]. *SLC20A2* encoding Pi transporter 2 (PiT2) has been reported as a causative gene. This mutated transporter was predicted to be unable to transport Pi from the extracellular environment [9,10]. We previously reported that Pi levels in the cerebral spinal fluid of IBGC patients with *SLC20A2* variants were significantly higher than in healthy controls [11]. These results suggest that defects in Pi homeostasis results in ectopic calcification even in the brain.

The formation of ectopic calcification is associated with various other factors such as oxidative stress, endoplasmic reticulum stress, increased apoptosis, increased DNA damage response, and decreased calcification regulator [12]. However, the mechanism by which a high concentration of Pi loading causes calcification is not fully understood. The main calcification component is thought to be calcium-Pi deposition [13,14,15]. The mineralization process is similar to bone formation and is due to transdifferentiation into osteoblast-like cells by an oversupply of Pi influx [16,17,18]. Indeed, calcification has been reported to occur in vascular smooth muscle cells (VSMCs) when Pi is excessive. In addition, inflammatory factor tumor necrosis factor (TNF)-α, a known risk factor for CKD, and bone morphogenetic protein (BMP)-2, an osteogenic factor, reportedly induce calcification by promoting phenotypic changes of VSMCs [19,20,21]. It is also known that inflammatory cells such as macrophages and T lymphocytes are involved in the formation of vascular calcification. Macrophages promote calcification by activating the osteogenic differentiation in VSMCs through the release of cytokines such as TNF-α and interleukin-6 (IL-6) [22,23]. Among cytokines produced and secreted by inflammatory cells, interferon-γ (IFN-γ) and TNF-α have been shown to induce Alkaline Phosphatase (ALP) expression in VSMCs [24]. Furthermore, TNF-α has been shown to promote osteoblast differentiation through induction of transcription factors such as runt-related transcription factor 2 (Runx2) and muscle segment homeobox 2 (Msx2) [25,26]. On the other hand, in the inflammatory reaction of the blood vessel wall, reactive oxygen species (ROS) is generated by the action of NADPH oxidase (NOX). It is known that ROS promotes apoptosis of VSMCs and differentiation into osteoblasts directly or through the production of oxidized low-density lipoprotein (LDL). Elucidating and controlling the mechanism of VSMCs phenotypic changes are considered useful for preventing vascular calcification and developing therapeutic strategies.

To further elucidate the vascular calcification mechanism, understanding the involvement of ROS, induced by a high concentration of Pi, is essential. ROS are produced during the general absorption and metabolism processes, but excess ROS accumulation causes various disorders in cells and tissues [27,28]. Indeed, previous studies have reported that ROS are induced by high Pi concentrations that are associated with vascular calcification [29,30,31]. ROS also inhibit the action of nitric oxide (NO) in vascular endothelial cells, thereby suppressing vasodilation and cell proliferation in VSMCs. ROS are also involved in the onset and progression of arteriosclerosis by inducing calcification [32]. Therefore, suppressing ROS may be a therapeutic strategy to treat vascular calcification induced by a high concentration of Pi loading.

Organic *gem*-dihydroperoxides (DHPs) are known to exert various beneficial effects [33,34,35]. Recently, a method for generating new DHPs that can be easily adjusted from commercially available compounds was reported [36,37]. Among these DHPs, 12AC3O leaded to apoptosis in K562 leukemia cells by scavenging intracellular ROS, without affecting the growth of peripheral blood monocytes (PBMCs) and fibroblasts [38]. In a previous study, we also found that the DHP 12AC2O exerted a neuroprotective effect by inhibiting abnormal protein accumulation via the direct trapping of intracellular ROS in amyotrophic lateral sclerosis model cells [39]. These results suggest that DHPs act as ROS scavengers to reduce intracellular ROS damage. In the present study, we investigated the effect of 12AC3O against high concentrations of Pi-induced calcification in VSMCs.

## 2. Results

### 2.1. Selection of DHPs and Cell Toxicity Assay

According to previous studies, we selected the 12AC series of DHPs [38]. Among the 12AC series of DHPs, 12AC3O was selected in anticipation of an antioxidative effect since 12AC3O showed the most anti-cancer activity (Figure 1A). 12AC3O’s cytotoxicity was confirmed using the mouse aortic smooth muscle cell line p53LMAco1. This cell line is a pure aortic SMC line, derived from a p53 knockout mouse, and sustains several differentiated characters of adult SMCs [40,41]. Previously, it was confirmed that 12AC3O did not show cytotoxicity in PBMCs, ASF-4-1 cells, and MCF10A cells [38]. Similarly, no cytotoxicity was observed when P53LMAco1 cells were treated with 1.0–10.0 µM of 12AC3O for 24 h (Figure 1B). We also examined the Pi transport activity in p53LMAco1 cells. The results showed that there was no change in Pi transport activity when cells were treated with 0.1– 3.0 µM of 12AC3O for 24 h (Figure 1C). These results showed that 12AC3O was not cytotoxic or associated with Pi transport activity under our experimental conditions.

### 2.2. Establishment of an Experimental Model of High Concentrations of Pi-Induced Calcification in p53LMAco1 Cells

Several reports showed that high concentrations of Pi-induced calcification in VSMCs [16,17,18]. Alizarin red and von Kossa staining were used to investigate, semi-quantitatively, the calcification of VSMCs. First, we examined the treatment time and concentration of Pi to determine the optimal conditions under which p53LMAco1 cells showed calcification by Pi loading. Pi (1.0 mM) was used as a control. As a result, Alizarin red staining showed that p53LMAco1 cells exhibited calcification after treatment with Pi concentrations higher than 3.0 mM (Figure 2A,B). In addition, p53LMAco1 cells significantly calcified by Pi loading with 3.0 mM for more than 4 days (Figure 2E,F). Similar results were obtained for von Kossa staining (Figure 2C,D,G,H). Based on the above results, 3.0 mM Pi treatment for 4 days was established as a calcification model of high concentrations of Pi loading in p53LMAco1 cells.

### 2.3. Antioxidants Suppress High Concentrations of Pi-Induced Calcification

A previous study reported that excess ROS, via NADPH oxidase, plays an important role in high concentrations of Pi-induced calcification [21]. Therefore, we used several antioxidants, including *N*- acetylcysteine (NAC, a foundational antioxidant) [42], edaravone (a free radical scavenger) [43,44], apocynin (an NADPH oxidase inhibitor) [45,46], and phosphonoformic acid (PFA, a Pi transporter inhibitor) [47], to investigate whether ROS were involved in the high concentrations of Pi-induced calcification of p53LMAco1 cells (Figure 3). The treatment concentrations of antioxidants and the Pi transporter inhibitor were determined based on previous studies. All antioxidants significantly suppressed high concentrations of Pi-induced calcification, as demonstrated by both Alizarin red and von Kossa staining (Figure 3A–D). Similarly, PFA completely inhibited the calcification. These results suggest that ROS play an important role in calcification involving high concentrations of Pi, and the removal of ROS helps suppress calcification in p53LMAco1 cells.

### 2.4. 12AC3O Suppresses High Concentrations of Pi-Induced Calcification

As mentioned above, antioxidants suppressed calcification, induced by a high concentration of Pi loading, suggesting that ROS are involved in the calcification of VSMCs. Thus, we confirmed the effect of 12AC3O against the calcification, induced by a high concentration of Pi loading in p53LMAco1 cells. Alizarin red staining showed that 12AC3O significantly inhibited the high Pi- induced calcification (Figure 4A,B). In addition, von Kossa staining showed that 12AC3O clearly inhibited the calcification (Figure 4C,D). These results suggest that 12AC3O suppresses calcification by reducing ROS.

Calcification is known to proceed in a process similar to bone formation, and several factors such as Msx2 and Runx2 involved in bone formation have been identified as important markers for calcification [12]. We examined changes in gene expression associated with calcification due to high concentration of Pi loading. It was confirmed that high concentration of Pi loading increased the gene expression of Runx2, a transcription factor associated with osteoblast differentiation (Figure 4E). 12AC3O significantly suppressed the expression of Runx2 gene (Figure 4F).

### 2.5. 12AC3O Suppresses High Concentrations of Pi-Induced Oxidative Stress

To further examine whether oxidative stress, caused by a high concentration of Pi loading, is attenuated by treatment with 12AC3O, redox-responsive dyes were used (Figure 5). CellROX and MitoSOX were used to evaluate intracellular oxidative stress and mitochondrial-specific oxidative stress, respectively. The CellRox signal was significantly increased by treatment with 3.0 mM Pi for 24 h. Simultaneous treatment with 3.0 mM Pi and 12AC3O clearly attenuated the CellROX signal (Figure 5A,B). Likewise, the signal of MitoSOX was obviously attenuate by the simultaneous treatment of 3.0 mM Pi and 12AC3O (Figure 5A,C).

### 2.6. 12AC3O Directly Traps Superoxide Anion and Hydroxyl Radical

Previous studies have shown that DHPs directly trap ROS [38,39]. To determine whether 12AC3O directly captures ROS, we performed an ESR analysis (Figure 6). ESR analysis showed that 12AC3O directly scavenged superoxide anion (Figure 6A,B) and hydroxyl radical (Figure 6C,D). These results suggest that the 12AC3O-mediated inhibition of high concentrations of Pi-induced calcification clearly involves its antioxidant ability to scavenge ROS.

## 3. Discussion

Ecotopic calcification is a high prevalent vascular phenotype that has associated with aging, atherothrombotic cardiovascular disease, diabetes mellitus, and CKD. It is known that ecotopic calcification is promoted by transdifferentiation of VSMCs. Complex VSMCs biology important in the pathogenesis of atherosclerosis, however, remains poorly understood. Although the pathogenesis of calcification is not known, hypoxia signaling in the cardiovascular system has been revealed as an upper activation pathway in VSMC in myocardial infraction patients [48,49].

The calcification pathways, caused by a high concentration of Pi, have been gradually elucidated. VSMCs exposed to procalcifying levels of phosphate, akin to what may occur in patients with CKD, lose expression of the smooth muscle contractile proteins SM22α and SM α-actin and express the bone markers Runx2, osteopontin, osteocalcin, and alkaline phosphatase [50]. The incorporation of excess extracellular Pi into cells, mainly through the Pi transporter PiT1, leads to ROS production [51]. Oxidative stress has been implicated in vascular calcification and shown to promote SMC differentiation. Increased activity of NADPH oxidase and elevated levels of hydrogen peroxide initiate SMC differentiation by upregulating Runx2 expression [12]. The disruption of Pi homeostasis causes ROS dysregulation via NADPH and mitochondria dysfunction [21,52]. Several studies have reported the relationship between calcification and ROS [29,30,31]. These studies revealed that ROS promote calcification by inducing the differentiation to osteoblasts via the activation of Protein Kinase B (AKT) signaling [29]. In addition, nuclear factor-kappa B (NF-kB) [53,54] and mitogen-activated protein kinase (MAPK) [55] signaling pathways have been shown to be involved in calcification. Downstream transcription factors of the MAPK signaling pathway, including Runx2 and Msx2, play an important role in the calcification of VSMCs [22,56].

Furthermore, ROS are known to accumulate with aging. Since accumulated ROS increases calcification, risk factors, such as CKD, leading to calcification include aging [57,58]. In the present study, high concentrations of Pi-induced calcification by increasing ROS production in VMSCs, although the detailed mechanisms remain to be elucidated. Additionally, 12AC3O decreased ROS by heavy loading of Pi and subsequently inhibited calcification. Thus, antioxidants could prevent calcification by inhibiting ROS, and 12AC3O (or its derivative) may be a candidate for developing agents to prevent calcification.

IBGC is a rare genetic disease, characterized by symmetric calcification in the basal ganglia and other brain regions. Previously, *SLC20A2*, which encodes a Pi transporter, has been identified as the causative gene for IBGC [9]. In addition, *XPR1*, another gene encoding a Pi transporter, has been identified. Therefore, it is presumed that the high concentration of Pi surrounding blood vessels contributes to ectopic calcification in the pathological condition of IBGC. Subsequently, disrupted Pi homeostasis inside and outside cells is expected to occur in IBGC. In our previous study, we demonstrated that mitochondrial-related ROS were produced by the disruption of Pi homeostasis [59]. In the present study, we showed that 12AC3O reduced the signal of Mito SOX, which evaluates mitochondrial oxidative stress. Therefore, 12AC3O may show potential beneficial effects on vascular calcification in IBGC.

12AC3O is new compound which can be easily synthesized. In this research, we showed that 12AC3O has the effect of suppressing oxidative stress similar to other antioxidants and directly scavenging oxidative stress. However, the inhibitory effect of 12AC3O against calcification may be insufficient as a fundamental strategy for treating vascular calcification. The pathological cascade gradually progresses as long as the high phosphate state is maintained. Endoplasmic reticulum stress and autophagy are also involved in the vascular calcification process [60]. It is also known that oxidative stress induces autophagy to protect cells. It is necessary to continue to investigate how the ROS levels produced by a high concentration of Pi loading affect cellular functions. Indeed, the reduction of Pi levels in vivo is essential for suppressing calcification.

In conclusion, 12AC3O, one of the novel DHPs, inhibited high Pi-induced calcification in VSMCs. Additionally, 12AC3O obviously attenuated Pi-induced ROS and exerted a direct scavenging activity against superoxide anion and hydroxyl radical. It has been reported that various pathways are involved in calcification, caused by a high concentration of Pi loading, and ROS production is an important factor. Therefore, 12AC3O may effectively inhibit the formation of pathological calcifications. These results support 12AC3O as a potential candidate to attenuate ROS-related ectopic calcification.

## 4. Materials and Methods

### 4.1. Cell Culture

The P53LMACO1 cell line (JCRB0150) used in this study was purchased from (Japanese Collection of Research Bioresources Cell Bank (JCRB Cell Bank, Osaka, Japan). p53LMAco1 cells were cultured in Dulbecco’s modified Eagle’s medium (DMEM) supplemented with 10% (*v/v*) fetal bovine serum (FBS) and maintained at 37 °C in humidified 5% CO_2_/95% air. For the high Pi experiment, p53LMAco1 cells were seeded on a 12-well plate in DMEM containing 10% FBS for 24 h. The medium was then replaced with 10% FBS-DMEM containing the appropriate amounts of sodium phosphate buffer (0.1 M Na_2_HPO_4_/NaH_2_PO_4_, pH 7.4) to produce final Pi concentrations of 3.0 mM. Ten percent FBS-DMEM medium was used as the control in this study.

### 4.2. Neurotoxicity Assays

p53LMAco1 cells were seeded at a density of 5.0 × 10^3^ cells/well in 96-well plates in DMEM containing 10% FBS. p53LMAco1 cells were incubated with or without 12AC3O (1.0, 3.0, or 10 µM) for 24 h. Triton-X was used as a negative control. The amount of LDH was measured in the culture supernatant. Cell toxicity was measured using an LDH assay kit following the protocol (Wako Pure Chemical Industries Ltd., Osaka, Japan).

### 4.3. ^32^Pi Transport Assays

A Pi uptake assay was performed using 3.0 mM Pi- and 12AC3O-treated cells grown to confluency in 24-well plastic plates, as previously described. The transport rate was expressed as nmol Pi per minute per mg protein [61].

### 4.4. qRT-PCR

Cells were seeded at the density of 1.0 × 10^5^ cells/well in a 12-well multiplate and cultured for 24 h. After that, p53LMAco1 was treated with 1.0 mM Pi as control or 3.0 mM Pi as vehicle and 12AC3O (0.1, 1.0, 3.0 µM) for 24 h. Total RNA was extracted using TriPure Isolation reagent (Sigma, St. Louis, MO, USA), following the manufacturer’s protocols. cDNA was prepared with ReverTra Ace^®^ qPCR RT Master Mix (Toyobo, Osaka, Japan) from 0.5 µg of total RNA, following the manufacturer’s protocols. An aliquot of diluted cDNA was applied to qRT-PCR. qRT-PCR analysis was performed using THUNDERBIRD^®^ SYBR qPCR Mix (Toyobo) and amplified using a StepOne Real-Time PCR System (Thermo Fisher Scientific Inc., Waltham, MA, USA). Primers used in the real time RT-PCR analysis were: β-actin (forward: 5′- GGCCAACCGGGAGAAAA-3′, reverse: 5′-GAGGCATAGAGGGACAGCACA-3′), Runx2 (forward: 5′-TGCAAGCAGTATTTACAACAGAGG-3′; reverse: 5′-GGCTCACGTCGCTCATCTT-3′). The level of β-actin cDNA in the sample was used as an internal control for all PCR amplification reactions.

### 4.5. Alizarin Red Staining

Alizarin S (100 mg) was dissolved in 10 mL of purified water and adjusted to pH 6.4 using a 0.1% KOH solution. Previously treated cells were washed with 1×Phosphate Buffered Saline (PBS), fixed with 4% paraformaldehyde, washed with purified water, and then stained with alizarin solution for 10 min. After washing three times with purified water, the sample was observed with an all-in-one fluorescence microscope, and image analysis was performed using ImageJ (NIH, Bethesda, MD, USA).

### 4.6. Von Kossa Staining

Treated cells were washed with 1×PBS, fixed with 4% paraformaldehyde, washed with purified water, treated with 5% aqueous silver nitrate solution, and incubated at room temperature for 2 h. UV (365 nM) was irradiated for 30 min on a Benchtop 2UV Transilluminators (UVP) (Analytik Jena, Jena, Germany). After washing with purified water, a 5% aqueous sodium thiosulfate solution was added over 3 min. After washing with purified water, the sample was observed with a fluorescence microscope, and image analysis was performed using ImageJ (NIH).

### 4.7. ROS Detection

To detect intracellular ROS production, we used the redox-sensitive dyes CellROX Green (Thermo Fisher Scientific Inc.) and MitoSOX Red (Thermo Fisher Scientific Inc.). CellROS was used to detect intracellular ROS, and MitoSOX was used to detect mitochondrial superoxide. After p53LMAco1 cells were prepared in uncoated glass-bottomed microwells, CellROX was added to the medium to a final concentration of 2.5 µM, and MitoSOX was added to the medium to a final concentration of 5 µM. Cells were incubated with CellROX or MitoSOX at 37 °C for 30 min or 10 min, respectively. Treated cells were fixed with 4% paraformaldehyde. After fixation, nuclei were stained with 4′,6-diamidino-2-phenylindole (DAPI) (1:2000), and then cells were encapsulated with VECTASHIELD Mounting Medium (Vector Laboratories, Burlingame, CA, USA). The sample was observed and photographed with a fluorescence microscope (Zeiss LSM 700, Carl Zeiss, Oberkochen, Germany), and fluorescence intensity was measured using Image J (NIH).

### 4.8. ESR Analysis

The ESR analysis (JES-FA 200, JEOL) was performed as The ESR analysis (JES-FA 200, JEOL) was performed as described previously [39].

### 4.9. Statistics

Data were presented as the mean ± S.E.M. The significance of differences was determined by an analysis of variance. Further statistical analysis for post-hoc comparisons was performed using the Bonferroni/Dunn test (SigmaPlot 11, Systat Software Inc., San Jose, CA, USA).

## Figures and Tables

**Figure 1 ijms-21-04628-f001:**
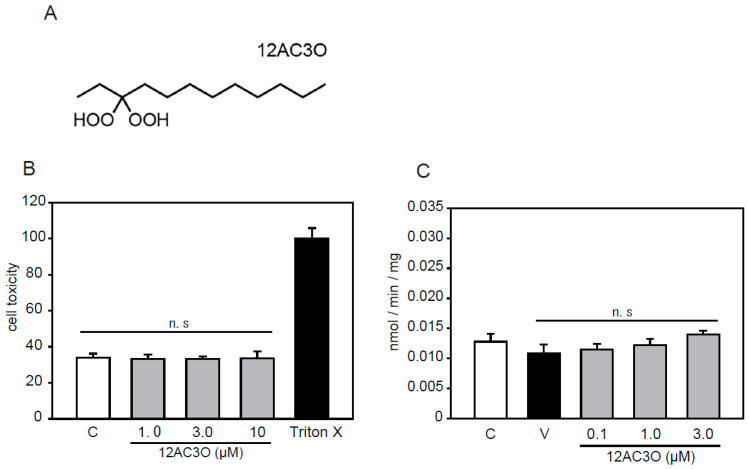
*gem*-Dihydroperoxide (DHP) selection and cytotoxicity evaluation. (**A**) Chemical structures of 12AC3O. (**B**) p53LMAco1 cells were cultured with Fetal Bovine Serum (FBS) -free medium in the presence or absence of DHPs (1.0, 3.0, and 10 µM for 24 h). The cytotoxicity of 12AC3O was measured using the lactate dehydrogenase (LDH) assay. Triton-X was used as a maximum LDH activity control. Results are presented as the mean ± standard error of the mean (S.E.M) of eight independent experiments. (**C**) The phosphate (Pi) uptake assay was performed in p53LMAco1 cells treated with Pi and 12AC3O for 24 h. C: Control (1.0 mM Pi), V: Vehicle (3.0 mM Pi). n.s: not significant.

**Figure 2 ijms-21-04628-f002:**
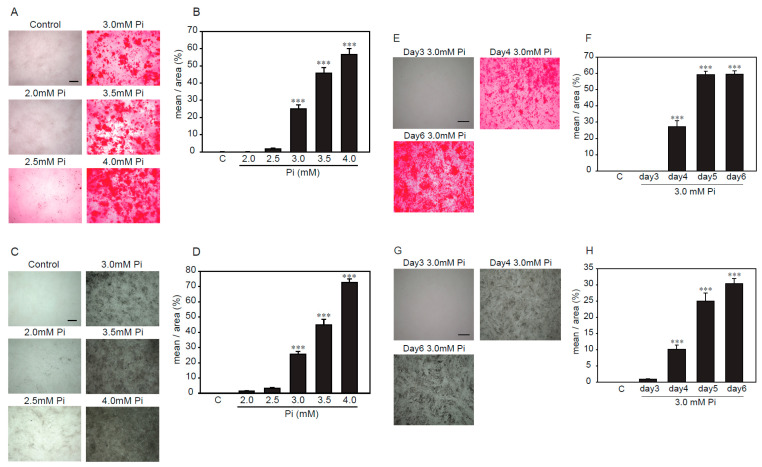
Determination of calcification conditions in p53LMAco1 cells. (**A**–**D**) p53LMACO1 cells were cultured under high concentrations of Pi (2.0–4.0 mM) for 6 days. (**E**–**H**) p53LMACO1 cells were cultured with 3.0 mM Pi for 3–6 days. Alizarin red staining (**A**,**B**,**E**,**F**) and von Kossa staining (**C**,**D**,**G**,**H**) were used to semi-quantitatively investigate calcification. C: Control (1.0 mM Pi), ****p* < 0.001 vs. Control Scale bar: 500 µm.

**Figure 3 ijms-21-04628-f003:**
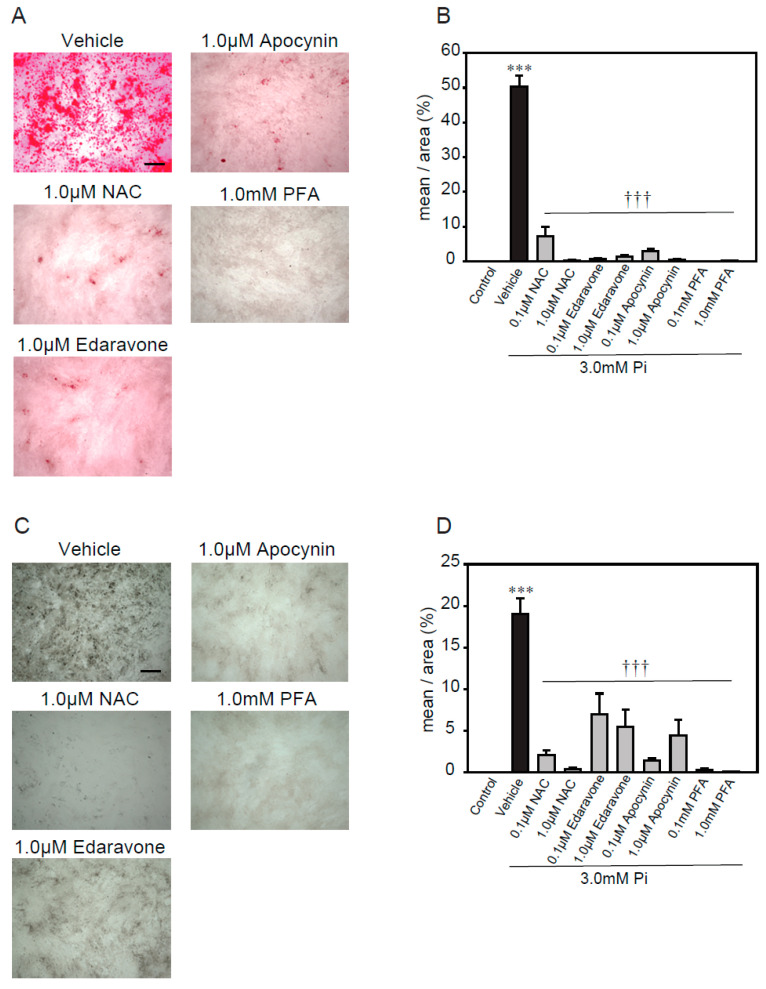
Antioxidants’ protective effects against high Pi-induced calcification. p53LMACO1 cells were cultured with 1.0 mM Pi or FBS (+) medium as a control and 3.0 mM Pi or FBS (+) medium in the presence or absence of various antioxidants for 4 days. (**A**,**B**) Alizarin red staining. (**C**,**D**) von Kossa staining. Results are presented as the mean ± S.E.M of five independent experiments. *** *p* < 0.001 vs. Control, ††† *p* < 0.001 vs. Vehicle. Scale bar: 500 µm.

**Figure 4 ijms-21-04628-f004:**
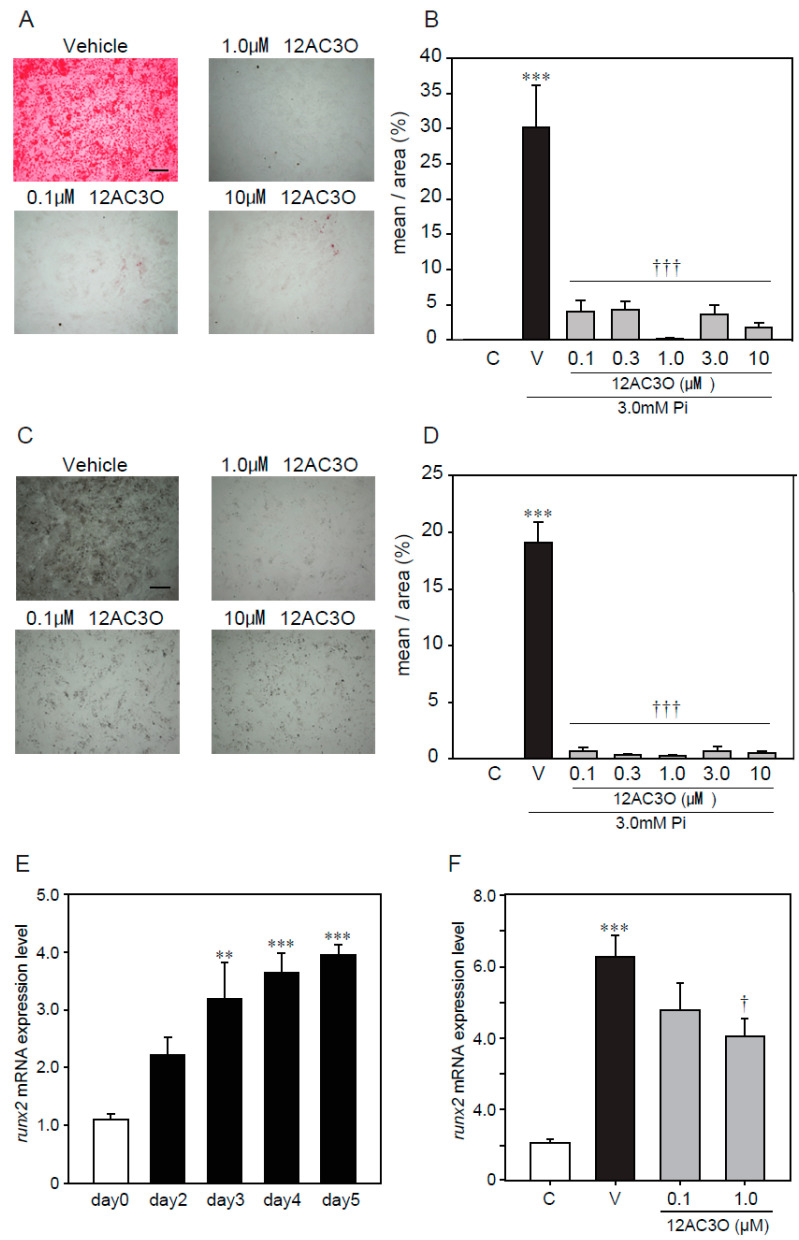
Protective effects of 12C3O against high Pi-induced calcification. p53 LMACO1 cells were cultured with 1.0 mM Pi or FBS (+) medium as a control and 3.0 mM Pi or FBS (+) medium in the presence or absence of 12AC3O (0.1–10 µM) for 4 days. (**A**,**B**) Alizarin red staining. (**C**,**D**) von Kossa staining. (**E**) qRT-PCR analysis of the effects of Pi loading with 3.0 mM on the mRNA expression level of Runx2. (**F**) p53LMAco1 cells were treated with 3.0 mM Pi and 12AC3O (0.1 µM, 1.0 µM, and 3.0 µM) for 5 days. Runx2 mRNA expression level was measured by qRT-PCR. Results are presented as the mean ± S.E.M. of five independent experiments (**A**–**D**) and three independent experiments (**E**,**F**). C: Control (1.0 mM Pi), V: Vehicle (3.0 mM Pi), ** *p* < 0.01, *** *p* < 0.001 vs. Control, † *p* < 0.05 vs. Vehicle, ††† *p* < 0.001 vs. Vehicle. Scale bar: 500 µm.

**Figure 5 ijms-21-04628-f005:**
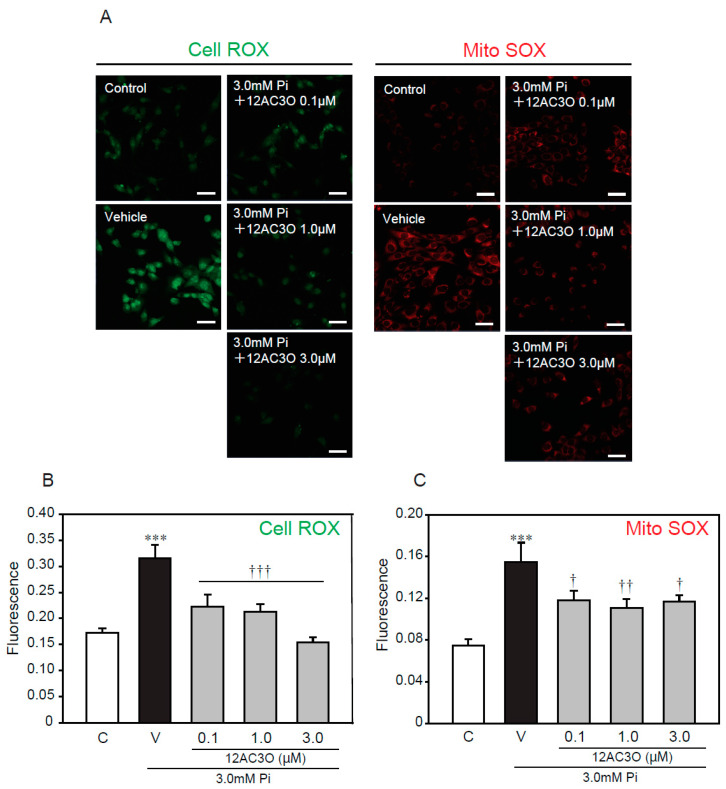
12AC3O suppresses high concentration of Pi-induced oxidative stress. (**A**–**C**) p53LMAco1 cells were treated with 3.0 mM Pi and 12AC3O for 24 h. The relative fluorescence intensity of Cell ROX (**A**,**B**) and Mito SOX (**A**,**C**) was measured with Image J. Results are presented as the mean ± S.E.M of seven independent experiments based on the fluorescence intensity of the control. C: Control (1.0 mM Pi), V: Vehicle (3.0 mM Pi). *** *p* < 0.001 vs. Control, ††† *p* < 0.001, †† *p* < 0.01, † *p* < 0.05 vs. Vehicle. Scale bar: 100 μm.

**Figure 6 ijms-21-04628-f006:**
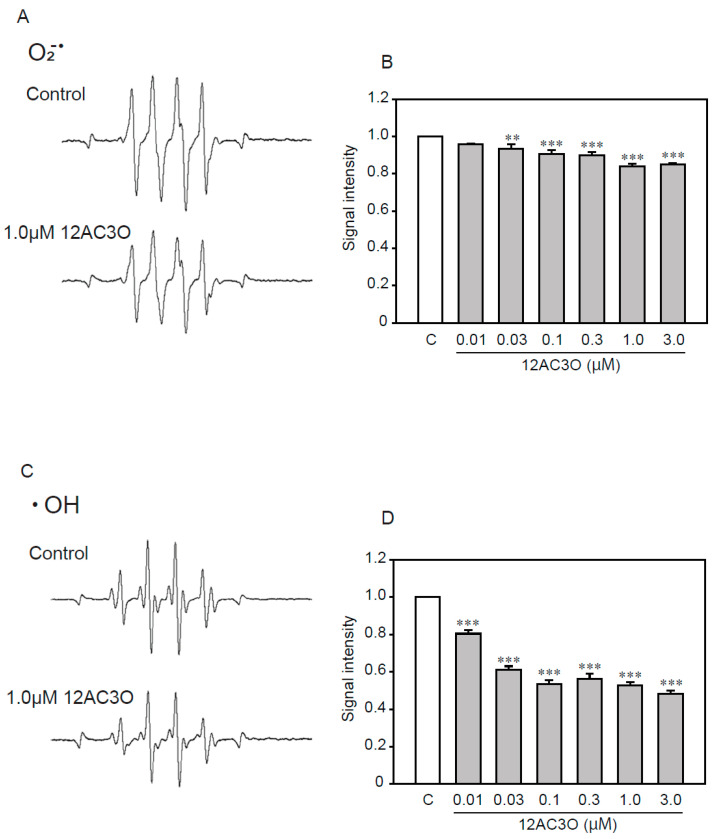
The direct trapping effect of 12AC3O against superoxide anion and hydroxyl radical. (**A**) Typical spectra of DMPO-OOH spin adduct. (**B**) Semi-quantitative measurement of DMPO-OOH spin adduct. (**C**) Typical spectra of DMPO-OH spin adduct. (**D**) Semi-quantitative measurement of DMPO-OH spin adduct. Results are presented as the mean ± S.E.M of four determinations based on the control. C: Control. ** *p* < 0.01, *** *p* < 0.001.

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
