# Peer review of "The Novel gem-Dihydroperoxide 12AC3O Suppresses High Phosphate-Induced Calcification via Antioxidant Effects in p53LMAco1 Smooth Muscle Cells"

_ijms, 2020, doi:10.3390/ijms21134628_

Round 1
Reviewer 1 Report
Dear Author,
The novel gem-dihydroperoxide 12AC3O suppresses high phosphate-induced calcification via antioxidant effects in p53LMAco1 smooth muscle cells by Naoko Takase et al.
Manuscript focused on the idea that high concentration phosphate (intake or CKD related) eventually lead to calcification via reactive oxygen species (ROS). Author propose that ROS suppression may be a useful treatment strategy of vascular calcification. Also research team evaluated antioxidant effect of 12AC3O and ability to inhibited calcification through antioxidative capacity Importantly, 12AC3O also attenuated oxidative stress. Research team use vascular mouse aortic smooth muscle cell line p53LMAco1. Reach team show: 1.Pi-induced vascular calcification and that 12AC3O inhibits vascular calcification by directly 12AC3O trapping. 2.ROS directly especially superoxide anion and hydroxyl radical.
Manuscript is well written. Experimental part is well described and clear. Team use appropriate methodology. Author used appropriate test to evaluate related changes in VSMCs. I would like to know why team specifically use this cell line p53LMAco1. It might not be wrong, but worth to be explained in the text. Also knocked p53 cells might be pro apoptotic with early senescent characteristic. This phenotype has elevated ROS level not related to phosphate
Result well-presented and convincing.
In the discussion, I would recommend to add abstract that shows complexity of ROS formation in atherosclerosis prompt cells. Recent publication presenting interesting fact that single ischemic event (MI) could alter ROS species in the vessel wall.:
Distinctive molecular signature and activated signaling pathways in aortic smooth muscle cells of patients with myocardial infarction
T Wongsurawat, et all
Atherosclerosis 271, 237-244
Transcriptome alterations of vascular smooth muscle cells in aortic wall of myocardial infarction patients
T Wongsurawat. et all
Data in brief 17, 1112-1135
Gene expression profile analysis of aortic vascular smooth muscle cells reveals upregulation of cadherin genes in myocardial infarction patients
AA Derda, et all
Physiological genomics 50 (8), 648-657
In the setting of CKD, environment even more complex Recent publication shows that interplay of many factors could affect VSMC transcript and play important role in SMC remodeling.
Role of Serpina3 in vascular biology
CC Woo
International journal of cardiology 304, 154-155
Also, author should mention that VSMCs not the only source of osteoblast- like cells in the arterial wall patients with CKD:
Vascular Calcification: Mechanisms of Vascular Smooth Muscle Cell Calcification.
Leopold JA.
Trends Cardiovasc Med 2015;25:267-74
Author Response
Dear Reviwer #1
We thank the Reviewer’s positive comments and thoughtful critiques and questions.
Manuscript is well written. Experimental part is well described and clear. Team use appropriate methodology. Author used appropriate test to evaluate related changes in VSMCs. I would like to know why team specifically use this cell line p53LMAco1. It might not be wrong, but worth to be explained in the text. Also knocked p53 cells might be pro apoptotic with early senescent characteristic. This phenotype has elevated ROS level not related to phosphate
Result well-presented and convincing.
In the discussion, I would recommend to add abstract that shows complexity of ROS formation in atherosclerosis prompt cells. Recent publication presenting interesting fact that single ischemic event (MI) could alter ROS species in the vessel wall.:
Distinctive molecular signature and activated signaling pathways in aortic smooth muscle cells of patients with myocardial infarction
T Wongsurawat, et all
Atherosclerosis 271, 237-244
Transcriptome alterations of vascular smooth muscle cells in aortic wall of myocardial infarction patients
T Wongsurawat. et all
Data in brief 17, 1112-1135
Gene expression profile analysis of aortic vascular smooth muscle cells reveals upregulation of cadherin genes in myocardial infarction patients
AA Derda, et all
Physiological genomics 50 (8), 648-657
In the setting of CKD, environment even more complex Recent publication shows that interplay of many factors could affect VSMC transcript and play important role in SMC remodeling.
Role of Serpina3 in vascular biology
CC Woo
International journal of cardiology 304, 154-155
Also, author should mention that VSMCs not the only source of osteoblast- like cells in the arterial wall patients with CKD:
Vascular Calcification: Mechanisms of Vascular Smooth Muscle Cell Calcification.
Leopold JA.
Trends Cardiovasc Med 2015;25:267-74
Reply – According to your comment, we added the detail about that why we specifically use p53LMAco1 cell line (Page 3, Line 99 – Line 101). In the future, we would like to study effect of 12AC3O on vascular calcification under high Pi conditions using vascular smooth muscle cell models differentiated from iPS cells and animal models. In addition, we cited your recommend papers in the discussion.
Reviewer 2 Report
Peer-review for manuscript entitled „The novel gem-dihydroperoxide 12AC3O suppresses high phosphate-induced calcification via antioxidant effects in p53LMAco1 smooth muscle cells
Brief summary
The aim of the paper was to analyze how the novel gem-dihydroperoxide 12AC3O affects the calcification of p53LMAco1 smooth muscle cells. The authors showed that 12AC3O inhibited high phosphate (Pi)-induced calcification of p53LMAco1 cells by scavenging ROS (superoxide anion and hydroxyl radical) and attenuating oxidative stress.
Broad comments
Strengths
The authors demonstrated that 12AC3O has a potent ROS-scavenging potential which might be a useful tool to decrease high Pi-induced vascular calcification. They showed that the beneficial effect of 12AC3O is attributed to its superoxide and hydroxyl radical trapping activity.
Weakness
The methodology of the manuscript is weak in its present form, more molecular biological methods (quantitave real-time PCR and immunoblot) must be performed to support the data.
Materials and methods section
Line 236 Toxicity assay In my opinion, Triton X cannot be called as negative control. It is a Triton X control which kills all the cells resulting in 100% cell death. It can be called as maximum LDH activity control.
Results section
Fig 1B What is C? This should be described in the figure legend,
Fig 1C What is C and V? These should be described in the figure legend. Based on the data of Fig 3, V is presumably vehicle that refers to phosphate (Pi). This should be clarified. In the later figures, the authors measured calcification after 4 days. If Fig 1C represents intracellular Pi level, it should be measured after 4 days as was performed for Alizarin Red and von Kossa staining. It is unlikely that cells accumulate detectable Pi after 1-day-treatment. Besides, many commercial kits are available that easily detect intracellular Pi levels. I suggest to measure intracellular Pi level with a commercial kit after several days, preferably after 4 days.
Fig 3 Antioxidants are well-known inhibitors of vascular calcification PMID: 31754473. The authors made a considerable effort to test several antioxidants. I suggest to combine Fig 3 and Fig 4 to highlight the effectivity of compound 12AC3O in comparison with well-known antioxidants. Moreover, it is necessary to support these data by molecular biological methods, i.e. by measuring the mRNA and protein levels of well-known markers of smooth muscle calcification such as Runx2, BMP-2, Msx-2, or analyzing the level of at least one smooth muscle cell marker (SM22α or α-actin)
Typesetting errors
How often were the culturing media changed during the experiments?
MM section line 234 dot is not needed after mM
Line 236 4.2 says ’Neurotoxicity assay’ It is a cytotoxicity assay in smooth muscle cells
Author Response
Dear Reviewer #2
We would appreciate the Reviewer’s very useful comments and thoughtful critiques and questions. Those comments are all valuable and very helpful for revising and improving our paper. We revised the manuscript based on your comments.
Comment 1 - Line 236 Toxicity assay In my opinion, Triton X cannot be called as negative control. It is a Triton X control which kills all the cells resulting in 100% cell death. It can be called as maximum LDH activity control.
Reply 1 – Thank you for pointing out our recognition mistake. It is correct to use the notation as you say. We corrected the manuscript (Page 4, Line 112 – Line 113).
Comment 2 - Fig 1B What is C? This should be described in the figure legend.
Fig 1C What is C and V? These should be described in the figure legend. Based on the data of Fig 3, V is presumably vehicle that refers to phosphate (Pi). This should be clarified.
Reply 2 – Thank you for your kind comment. According to your comment, we corrected the manuscript (Page 4, Line 115).
Comment 3 - In the later figures, the authors measured calcification after 4 days. If Fig 1C represents intracellular Pi level, it should be measured after 4 days as was performed for Alizarin Red and von Kossa staining. It is unlikely that cells accumulate detectable Pi after 1-day-treatment. Besides, many commercial kits are available that easily detect intracellular Pi levels. I suggest to measure intracellular Pi level with a commercial kit after several days, preferably after 4 days.
Reply 3 – We completely agree with your comment and understand the importance of measuring intracellular Pi levels. We also know that some commercial kits are available as you say. Unfortunately, it is difficult to properly measure the intracellular Pi levels under the conditions of our research. We will measure the accurate intracellular Pi levels using RI probe and so on.
Comment 4 - Fig 3 Antioxidants are well-known inhibitors of vascular calcification PMID: 31754473. The authors made a considerable effort to test several antioxidants. I suggest to combine Fig 3 and Fig 4 to highlight the effectivity of compound 12AC3O in comparison with well-known antioxidants.
Reply 4 – We agree with your comments. We should compare the anti-oxidative effect of 12AC3O and several anti-oxidants. We think 12AC3O is a lead compound for ROS-related ectopic calcification. In future, we are going to evaluate the effect of new 12AC3O derivatives according to your comments. Again, thank you for your very useful comments.
Comment 5 - Moreover, it is necessary to support these data by molecular biological methods, i.e. by measuring the mRNA and protein levels of well-known markers of smooth muscle calcification such as Runx2, BMP-2, Msx-2, or analyzing the level of at least one smooth muscle cell marker (SM22α or α-actin)
Reply 5 – According to your comments and Reviewer #3’s comment 1, we performed a qRT-PCR experiment using Runk2 primer set. We added new results (Page 6, Line 158 – Line 163, Figure 4 E and F) and Materials and Methods (Page 10, Line 276 – Page 11, Line 288).
Comment 6- Typesetting errors
How often were the culturing media changed during the experiments?
MM section line 234 dot is not needed after mM
Line 236 4.2 says ’Neurotoxicity assay’ It is a cytotoxicity assay in smooth muscle cells
Reply 6– Thank you for your kind comments. We corrected the manuscript.
Reviewer 3 Report
In the current study, Naoko Takase and group have studied the role of a compound, gem dihydroxyperoxide 12AC3O in diminishing the vascular calcification induced by Pi overload in vascular smooth muscle cells (VSMC). They observed that 12AC3O reduced calcification by trapping superoxide and hydroxyl ion, hence reducing ROS. The study is well designed and executed, however some of the concerns are as below.
1) Although the authors observed reduced calcification with 12AC3O in VSMC by alizarin and von kossa staining, to gain some mechanistic insight, it would be good to look at some of the important calcification markers like BMP-2, RUNX-2 and MSX-2 either mRNA or protein levels in their treated group.
2) It would be also good to look at Pit2 levels (mRNA or protein) with their 12 AC3O treatments which is of particular interest to authors and could help them gain some mechanistic insight
3) The authors should discuss some of the basic properties of 12AC30
4)The authors have mentioned about inflammation and BMP-2’s role in vascular calcification in the introduction, they could cite recent findings from Dube et al which shows a direct role of macrophages in vascular calcification by secreting BMP-2 and RUNX2. Also, they can briefly discuss the role of endothelium in vascular calcification too. See citations below. The endothelium and macrophages play an important role in vascular calcification in addition to VSMC, so its suggested to discuss them briefly in the introduction.
https://www.sciencedirect.com/science/article/abs/pii/S0006291X17314079
https://www.sciencedirect.com/science/article/abs/pii/S0021915017314442
https://onlinelibrary.wiley.com/doi/full/10.1111/joim.12605
5) The authors should make sure they include all the abbreviations in their list including ESR
6) minor grammatical and sentence formation check required
Author Response
Dear Reviewer #3
We appreciate reviewers’ interest in our work, and the helpful suggestions. Those comments are all valuable and very helpful for revising and improving our paper. We revised the manuscript based on your comments.
Comment 1 - Although the authors observed reduced calcification with 12AC3O in VSMC by alizarin and von kossa staining, to gain some mechanistic insight, it would be good to look at some of the important calcification markers like BMP-2, RUNX-2 and MSX-2 either mRNA or protein levels in their treated group.
Reply 1 – According to your comments and Reviewer #2’s comment 5, we performed a qRT-PCR experiment using Runk2 primer set. We added new results (Page 6, Line 158 – Line 163, Figure 4 E and F) and Materials and Methods (Page 10, Line 276 – Page 11, Line 288).
Comment 2 - It would be also good to look at Pit2 levels (mRNA or protein) with their 12 AC3O treatments which is of particular interest to authors and could help them gain some mechanistic insight.
Reply 2 – We thank the reviewer for this important suggestion. Several Pi transporters, including PiT2, are expressed in VSMCs. Previous reports suggest that PiT1, another Pi transporters, are involved in vascular calcification caused by a high concentration of Pi loading (Page 9, Line 209 – Line 211). However, the details of the regulational mechanism of Pi transporters remain unclear. From the 32Pi transport assays in the present study, at least 12AC3O had no effect on the amount of Pi uptake (Figure 1C). The effect of 12AC3O on the individual Pi transporters needs to be investigated further and will be the subject for further study.
Comment 3- The authors should discuss some of the basic properties of 12AC3O.
Reply 3– According to your comments, we added the basic properties of 12AC3O in discussion of the manuscript (Page 9, Line 236 – Line 238).
Comment 4- The authors have mentioned about inflammation and BMP-2’s role in vascular calcification in the introduction, they could cite recent findings from Dube et al which shows a direct role of macrophages in vascular calcification by secreting BMP-2 and RUNX2. Also, they can briefly discuss the role of endothelium in vascular calcification too. See citations below. The endothelium and macrophages play an important role in vascular calcification in addition to VSMC, so its suggested to discuss them briefly in the introduction.
Reply 4 – Thank you for letting us know the articles. We added these articles in the introduction of the manuscript (Page 2, Line 59 – Line 68).
Comment 5 - The authors should make sure they include all the abbreviations in their list including ESR.
Comment 6 - minor grammatical and sentence formation check required.
Reply 5&6 – Thank you for your kind comments. We corrected the manuscript.
Round 2
Reviewer 2 Report
Nice improvement .
Author Response
We'd appreciate your efforts and kindness.
Reviewer 3 Report
All questions addressed, thank you.
Author Response
Thank you very much for your wonderful advices.